# Exposure to *Aggregatibacter actinomycetemcomitans* before Symptom Onset and the Risk of Evolving to Rheumatoid Arthritis

**DOI:** 10.3390/jcm9061906

**Published:** 2020-06-18

**Authors:** Eduardo Gomez-Bañuelos, Linda Johansson, Maximilian F. Konig, Anders Lundquist, Merlin Paz, Kåre Buhlin, Anders Johansson, Solbritt Rantapää-Dahlqvist, Felipe Andrade

**Affiliations:** 1Division of Rheumatology, The Johns Hopkins University School of Medicine, Baltimore, MD 21224, USA; jgomezb1@jhmi.edu (E.G.-B.); konig@jhmi.edu (M.F.K.); mpaz5@jhmi.edu (M.P.); 2Department of Public Health and Clinical Medicine, Rheumatology, Umeå University, 901 85 Umeå, Sweden; linda.e.johansson@umu.se; 3Department of Statistics, USBE, Umeå University, 901 87 Umeå, Sweden; anders.lundquist@umu.se; 4Division of Periodontology, Department of Dental Medicine, Karolinska Institutet, 14104 Huddinge, Sweden; kare.buhlin@ki.se; 5Department of Odontology, Umeå University, 901 87 Umeå, Sweden; anders.p.johansson@umu.se

**Keywords:** rheumatoid arthritis, leukotoxin A, *Aggregatibacter actinomycetemcomitans*, Anti-citrullinated protein antibodies

## Abstract

Periodontal disease has been implicated in the pathogenesis of rheumatoid arthritis (RA), an autoimmune disease characterized by immune-mediated synovial damage, and antibodies to citrullinated antigens. Here, we investigate the association between exposure to the periodontal pathogen *Aggregatibacter actinomycetemcomitans* (*Aa*) and the development of RA. IgM, IgG and IgA antibodies to *Aa* leukotoxin A (LtxA) were detected by ELISA in plasma from a cohort of Swedish adults at different stages of RA development, from before onset of symptoms to established disease. Patients with early and established RA had increased levels of anti-LtxA IgM compared with matched non-RA controls and periodontally healthy individuals. Logistic regression revealed that anti-LtxA IgM levels were associated with RA during early disease (OR 1.012, 95%CI 1.007, 1.017), which was maintained after adjustment for smoking, anti-CCP antibodies, rheumatoid factor, *HLA-DRB1* shared epitope alleles and sex. We found no association between anti-LtxA IgG/IgA antibodies and RA at any stage of disease development. The data support a temporal association between anti-LtxA IgM antibodies and the development of RA, suggesting that a subset of RA patients may have been exposed to *Aa* around the time of transition from being asymptomatic to become a patient with RA.

## 1. Introduction

Although the pathogenesis of rheumatoid arthritis (RA) is not completely understood, it is acknowledged that the autoimmune response that leads to the initiation and propagation of the disease results from a complex interplay between genetic and environmental factors [1]. Periodontitis is a disorder induced by microbial dysbiosis causing loss of the tooth supporting tissues and bone [2]. Certain periodontal pathogens are considered potential triggers of autoimmunity in RA by inducing the production of citrullinated antigens, which are major targets of the autoantibody response in this disease [3]. In particular, *Aggregatibacter actinomycetemcomitans* (*Aa*), a Gram-negative bacterium associated with severe forms of periodontal disease [3,4], has a unique ability to induce the cellular repertoire of citrullinated antigens found in the RA joint [5].

*Aa* secretes leukotoxin A (LtxA), a pore-forming toxin that induces neutrophil death by a phenomenon termed leukotoxic hypercitrullination [6]. During this process, membranolytic damage by LtxA induces prominent intracellular calcium flux and osmotic lysis, leading to hyperactivation of citrullinating enzymes and hypercitrullination of a broad range of substrates [5,7]. IgG antibodies to LtxA used as markers of previous *Aa* exposure are significantly associated with established and early RA when compared to periodontally healthy (PH) subjects and healthy controls regardless of their periodontal status, respectively [5,8,9]. Together, these studies provide mechanistic and clinical evidence implicating *Aa* in the pathogenesis of RA.

In order to gain further insights into the potential role of *Aa* in RA pathogenesis, we examined the temporal association between *Aa* exposure and onset of RA by studying patients at different stages of disease development, from before onset of symptoms of RA to established disease.

## 2. Materials and Methods

### 2.1. Study Population

The study included the analysis of plasma samples from adult patients from Sweden classified as RA by the American Rheumatism Association (ARA) 1987 criteria [10] collected before the development of symptoms (pre-symptomatic individuals), during early disease (early RA, duration of symptoms <12 months) and established RA (duration of symptoms ≥5 years). Subjects were identified by co-analyzing the registers of the Medical Biobank cohorts and Rheumatology Department at the University Hospital of Umeå. The Medical Biobank cohorts are population-based health surveys to which all habitants of Västerbotten county (Sweden) are continuously invited to participate. A more detailed description of enrollment, blood sampling and sample storage has been described previously [11].

A total of 527 pre-symptomatic individuals were identified. The median (inter quartile range (IQR)) pre-dating time until the onset of RA symptoms was 5.2 (6.0) years. At least one sample was found for 526 pre-symptomatic individuals, two samples for 13 individuals and three samples for one individual. The early RA group included 231 patients all of whom had also donated a sample pre-dating symptom onset. Patients with established RA comprised 168 subjects (median (IQR) disease duration, 8.0 (3) years), including 108 who donated a sample during early RA. All patients were diagnosed at the Early Arthritis Clinic at University Hospital, Umeå. The control group was composed of 567 individuals without RA diagnosis (with unknown periodontal status), who were selected randomly from the Medical Biobank cohort, matched according to sex, year of birth and date of sample collection to the pre-symptomatic subjects. Smoking status was defined as ever being a smoker (either previous or current smoker) or a non-smoker. We also included 73 samples from PH individuals, defined as previously described [12], that were collected at the Department of Periodontology, University Hospital, Huddinge, Stockholm. The Regional Ethics Committees at Umeå University (No. Dnr 2013-347-31M) and the Karolinska Institutet (No. 347/99 and 220/03), Sweden approved the study. All participants gave their informed consent before enrollment. The study was conducted according to the declaration of Helsinki. A summary of demographics of the study population are shown in Appendix A.

### 2.2. Detection of Anti-CCP, RF and HLA Genotyping

Anti-CCP IgG were measured in serum using the anti-CCP2 antibody test according to the manufacturer’s instructions (Euro Diagnostica, Malmö, Sweden). Rheumatoid factor (RF) IgM isotype was determined by EliA assay using the Phadia 2500-system (Phadia GmbH, Freiburg, Germany). Genotyping of HLA-SE was performed as previously described [13]. HLA-shared epitope (SE) was defined as HLA -DRB1*0101/0401/0404/0405/0408.

### 2.3. Detection of Anti-LtxA Antibodies

The coding sequence of the immunodominant C-terminal region (CTR, amino acids 730-1055) of LtxA was cloned from *Aa* SUNY ab75 into pET28a(+) [5]. Recombinant His-tagged LtxA-CTR was expressed in *E. coli* BL21 (DE3) and purified by Ni-NTA affinity chromatography. Serum IgG, IgA and IgM antibodies against purified LtxA-CTR were detected by ELISA. Briefly, poly-styrene plates (Nunc-Maxisorp) were coated overnight with 100 ng/well of purified LtxA-CTR in PBS pH 7.4, or PBS alone. Plates were blocked in PBS 0.1% Tween (PBST) plus 3% non-fat dry milk (PBST-M). Sera were diluted at 1:1000 in PBST-M 1% and assayed in duplicate, using antigen-coated wells and wells without antigen for background subtraction. HRP-conjugated anti-human IgG, IgA and IgM were used as a secondary antibody (diluted at PBST-M 1%; dilution was 1:20,000 for anti-human IgG and IgA, and 1:5000 for anti-human IgM). Anti-LtxA arbitrary units (AU/mL) were calculated using a serial dilution of a human sera with high levels of anti-LtxA IgG, IgA and IgM antibodies as standard curve. The cut-off value for anti-LtxA isotypes’ positivity was defined by using receiver operating characteristic curve (ROC), to achieve specificity of 95% for anti-LtxA IgA, and anti-LtxA IgG, with cut-off values of ≥ 58.21 AU/mL, and ≥ 45.23 AU/mL, respectively. The specificity for anti-LtxA IgM was set at 98% and ≥ 64.71 AU/mL was considered positive.

### 2.4. Statistical Analyses

Continuous data was compared with Student’s *t* test. The Chi-square test or Fisher’s exact test was used when analyzing categorical data. Logistic regression analyses with cases and controls as dependent variable were used to identify associations between LtxA antibody positivity or concentration and risk factors with the development of RA. Any adjustments of the data (e.g., for sex, age and smoking) were based on previous data or hypothesis. The associations are presented as odds ratios (OR) with 95% confidence interval (CI). *p* ≤ 0.05 was considered as statistically significant. To ensure comparability between anti-IgM antibody levels analyzed at separate occasions we standardized the observed values at each occasion to obtain z-scores and then multiplied the z-scores with the standard deviation of the reference occasion and added the mean value of the reference occasion. The steps ensure that the transformed values have the same mean and standard deviation in all three measurements. The statistical analyses were performed using SPSS 23.0 software (Chicago, IL, USA) and R (R version 3.4.3).

## 3. Results

### 3.1. IgM Antibodies Against LtxA are Associated with RA.

To establish the prevalence and significance of *Aa* exposure at different stages of RA development, we assayed IgM, IgA and IgG antibodies to LtxA in sera from 73 PH individuals, 567 non-RA controls, 526 pre-symptomatic individuals, 231 patients with early RA, and 170 patients with established RA. The levels and prevalence of the anti-LtxA IgM, IgG and IgA are summarized in Figure 1 and Table 1.

Anti-LtxA IgM antibodies were enriched in the RA groups (both in early and established RA) when compared to PH and controls (Figure 1A). Levels of anti-LtxA IgM were significantly higher in early RA and established RA than PH (mean 28.0 vs. 9.8, *p* = 0.01; 20.1 vs. 9.8, *p* = 0.004, for early RA vs. PH and established RA vs. PH, respectively) and controls (mean 28.0 vs. 13.4, *p* < 0.001, and 20.1 vs. 13.4, *p* = 0.02, for early RA vs. controls and RA vs controls, respectively) (Figure 1A). The level of anti-LtxA IgG antibodies was significantly lower in PH (mean 49.7) compared to controls (mean 121.7, *p* = 0.008), pre-symptomatic individuals (mean 103.8, *p* = 0.012), early RA (mean 124.0, *p* = 0.006) and established RA (mean 107.8, *p* = 0.015) (Figure 1B). There was no difference in anti-LtxA IgA antibody levels between the study groups (Figure 1C).

Using a cut-off value for anti-LtxA positivity defined by ROC, patients with early RA showed an increased frequency of anti-LtxA IgM antibodies when compared with PH (12.6% vs. 1.4%, respectively; *p* = 0.005) and controls (12.6% vs. 3.9%, respectively; *p* < 0.001) (Table 1). Moreover, the prevalence of anti-LtxA IgG positivity was significantly increased in controls, early RA, and established RA when compared with PH (38.1% vs. 24.7%, *p* = 0.029; 37.7% vs. 24.7%, *p* = 0.042; and 39.3% vs. 24.7%, *p* = 0.033; for control vs. PH, early RA vs. PH, and established RA vs. PH, respectively) (Table 1). The frequency of anti-LtxA IgA positivity showed no differences among the study groups (Table 1).

### 3.2. Early RA Patients Have Evidence of Recent Exposure to Aa

Using anti-LtxA IgM antibodies as a marker of recent exposure to *Aa* [14], we classified subjects according to positivity to anti-LtxA isotype in four subgroups: no exposure (subjects negative for the three anti-LtxA isotypes), distant exposure (anti-LtxA IgM negative, anti-LtxA IgG and/or IgA positive), recent exposure (anti-LtxA IgM positive and positive for anti-LtxA IgG and/or IgA), and new exposure (only positive for anti-LtxA IgM antibodies) (Table 2). Interestingly, the frequency of subjects with no exposure and distant exposure to *Aa* was not different between controls and the distinct RA subgroups (Table 2). Nevertheless, it is intriguing that, different to controls, the proportion of subjects with evidence of recent exposure to *Aa* (anti-LtxA IgM, IgG and/or IgA positive individuals) was more frequent during pre-symptomatic (OR 2.12 95%CI 1.04, 4.30), early RA (OR 3.88 95%CI 1.81, 8.26) and established RA (OR 2.90 95%CI 1.21, 6.91) (Table 2). Moreover, serologic evidence of new exposure to *Aa* (i.e., anti-LtxA IgM positive with anti-LtxA IgG and/or IgA negative) was more frequent during the period of early RA (OR 2.58 95%CI 1.09, 6.09) (Table 2). Further analysis of the trajectory of anti-LtxA IgM antibodies from before onset of symptoms to established disease showed that the levels of anti-LtxA IgM were similar to controls during most of the pre-symptomatic stage and increased above controls before the start of symptomatic RA (Appendix A).

### 3.3. Association between Exposure to Aa, Sex, Smoking, RF, Anti-CCP and HLA-SE

Next, we analyzed the association between *Aa* exposure and known risk factors for RA such as sex, smoking, RF, anti-CCP and HLA-SE (Table 3, Appendix A). Levels of anti-LtxA IgM were increased in men with early RA and in women with established RA when compared to their matched sex controls. Smoking was associated with increased anti-LtxA IgM in subjects with early and established RA in comparison with controls. Interestingly, HLA-SE positive pre-symptomatic individuals and early RA also had higher levels of anti-LtxA IgM when compared to HLA-SE positive controls. However, we found no significant associations between levels of anti-LtxA isotypes and anti-CCP in any of the study groups. Similarly, there was no significant association between the levels of anti-LtxA IgA or IgG and sex, smoking status, RF, and HLA-SE (Appendix A).

### 3.4. Association between Anti-LtxA IgM Antibodies and the Development of RA

The predictive value of anti-LtxA IgM for the development of RA was assessed using multiple logistic regression. In the unadjusted logistic regression, levels of anti-LtxA IgM were independently associated with an increased risk of symptomatic RA (both early and established RA) (Table 4). After adjusting for smoking, anti-CCP positivity, RF, HLA-SE or sex, anti-LtxA IgM levels and positivity remained associated with early RA (Table 4 and Appendix A). In established disease, however, only anti-LtxA IgM levels remained significantly associated with RA after adjusting for smoking, HLA-SE or sex (Table 4). In pre-symptomatic individuals, anti-LtxA IgM levels and positivity conferred an increased risk of disease after adjustment for HLA-SE and sex, respectively (Table 4, and Appendix A). The levels and positivity of anti-LtxA IgA and IgG were not associated with the development of RA, independently of adjusting for smoking status, anti-CCP positivity, RF, carriage of HLA-SE, or sex.

## 4. Discussion

The loss of tolerance to citrullinated antigens is an early event that precedes the onset of RA [15] and has focused attention on identifying potential factors that dysregulate citrullination in the pre-symptomatic phase of the disease. Periodontal disease and certain oral pathogens (such as *Aa* and *Porphyromonas gingivalis*) have been associated with the production of citrullinated antigens in RA [3]. Here, using a cohort of Swedish adults, we found that prior exposure to *Aa* (determined by the presence of anti-LtxA IgG and/or IgA antibodies) is not a risk factor associated with pre-symptomatic cases or diagnosed RA. Nevertheless, a subset of patients with RA showed a significant increase of IgM antibodies to LtxA (an early serological marker of *Aa* infection) [14], suggesting that recent exposure to *Aa* may be associated with the symptom onset of the disease. While these findings are intriguing, several features of this study may not apply to other populations.

Swedish adults have a high prevalence of antibodies to LtxA (i.e., up to 57% combining IgG, IgA and IgM isotypes), which is independent of the presence of periodontitis [16,17]. Indeed, compared with other populations, we confirmed a high frequency of anti-LtxA IgG positivity in controls and PH individuals from Sweden. In a recent study of healthy controls without known periodontal status from Leiden, Volkov et al. [8] reported a prevalence of 22% of anti-LtxA IgG positivity as compared to 38% of Swedish controls (also with unknown periodontal status). In PH subjects from the United States [5], we previously reported a prevalence of 11% for anti-LtxA IgG positivity, while the prevalence was more than double in PH individuals from Sweden (25%). Whether these findings may suggest an increased risk of or susceptibility to asymptomatic *Aa* exposure in the Swedish population is unclear. However, the high prevalence of anti-LtxA positivity among healthy Swedish adults makes it difficult to confirm previous associations between *Aa* exposure and RA [5,8]. For instance, although individuals at different stages of RA development showed a significant increase of anti-LtxA IgG compared to PH subjects, this difference was not significant when compared to controls of unknown periodontal status. This finding importantly contrasts with the striking association between anti-LtxA IgG and RA reported by Volkov et al. [8,9] when comparing RA with healthy controls and with patients with non-RA arthritis.

Different to IgG and IgA antibodies to *Aa*, which serve as markers of previous *Aa* exposure as well as *Aa*-associated periodontitis, elevated IgM antibodies to *Aa* are considered indicators of recent *Aa* infection and potential predictors of imminent periodontitis associated with this pathogen [14]. Interestingly, in the context of a unique population in which anti-LtxA IgG is not useful to distinguish RA from controls, our data suggest that anti-LtxA IgM antibodies may identify a subset of patients who have been recently colonized or infected with *Aa* either during the transition from pre-symptomatic phase to early stages of RA or to established disease. This subset likely includes newly exposed patients (i.e., anti-LtxA IgM positive, IgG/IgA negative) and individuals with recent exposure who have initiated an adaptive immune response to *Aa* (i.e., anti-LtxA IgM, IgG and IgA positive). Taken together, the data suggest that primary *Aa* infection late in life, but not during childhood, may be relevant to determine the transition to symptoms and the development of RA.

Using logistic regression analysis, we further confirmed that recent exposure to *Aa* is associated with early RA. These analyses also illustrate how risk factors may determine the risk for RA during the different stages of the development of the disease. During the pre-symptomatic period, most of the risk was determined by known RA risk factors, such as smoking, anti-CCP, RF and HLA-SE. When transitioning to the disease, however, anti-LtxA IgM was predictive of early RA independently of smoking, anti-CCP, RF, HLA-SE and sex. These findings support a time-dependent association between a recent exposure to *Aa* and RA. Interestingly, this association was more evident when using anti-LtxA IgM levels rather than positivity in the logistic regression model. Here, each AU/mL of anti-LtxA IgM increased the OR to develop RA by 1.01. Therefore, the risk of developing RA would be directly proportional to the anti-LtxA IgM levels. This finding may suggest that the degree of exposure to *Aa* (e.g., asymptomatic carrier vs. *Aa*-associated periodontitis) could be important for the development of symptomatic disease.

It is noteworthy that anti-LtxA IgM antibodies were predictive for being a pre-symptomatic individual and a patient with RA when adjusted for HLA-SE, supporting a possible interaction between *Aa* and HLA-SE alleles in determining the susceptibility to develop RA. A potential interaction between anti-LtxA IgM antibodies and anti-CCP is also suggested in patients with early RA, which may imply a role of *Aa* in enhancing the immune response to citrullinated proteins at early stages of the disease.

Our study has some limitations. Only few subjects (*n* = 118) had matched samples in all RA groups, affecting our capacity to accurately determine the trajectory of anti-LtxA antibodies. We have no information regarding the oral health of both RA patients and controls except for the PH free individuals in order to associate exposure to *Aa* and periodontal status. The study design does not allow us to establish causality between RA and *Aa*, but only the association between exposure to *Aa* and RA. Thus, it is also plausible that the increased exposure to *Aa* in early RA may be the result of an increase of oral colonization as a consequence of the inflammatory environment already set during pre-symptomatic stage.

In summary, the data suggest a time dependent association between recent exposure to *Aa* and the development of RA symptoms in a subset of adults with RA. These findings support a model in which environmental factors, such as recent exposure to disease-relevant pathogens, may determine the transition from long standing asymptomatic autoimmunity to clinical disease. However, in the absence of an environmental trigger, healthy individuals with RA-associated autoantibodies may never evolve to RA.

## Figures and Tables

**Figure 1 jcm-09-01906-f001:**
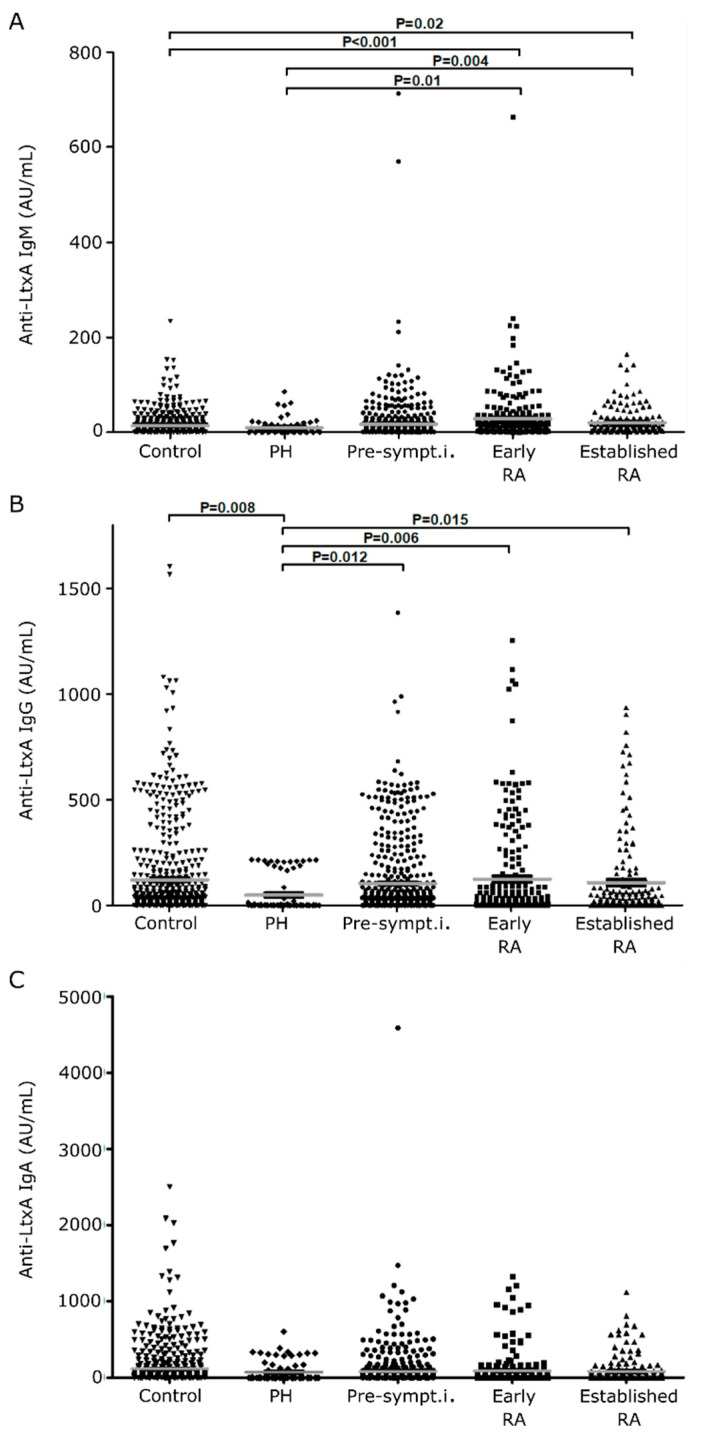
Levels of anti-LtxA antibodies according to isotype in the study groups. Levels of anti-LtxA IgM (**A**), anti-LtxA IgG (**B**) and anti-LtxA IgA (**C**) in controls, *n* = 567; periodontitis healthy individuals (PH), *n* = 73; pre-symptomatic individuals (pre-sympt.i.), *n* = 526; early RA, *n* = 231; and established RA, *n* = 168.

**Table 1 jcm-09-01906-t001:** Frequency of ever positive for anti-LtxA IgM, IgG and IgA in pre-symptomatic individuals, early rheumatoid arthritis (RA), established RA and periodontitis healthy individuals.

	Study Groups				
Anti-LtxA Isotype	PH *n* = 73	Control *n* = 567	Pre-sympt.i. *n* = 526	Early RA *n* = 231	Established RA *n* = 168
IgM positive, *n* (%)	1 (1.4)	22 (3.9)	33 (6.3)	29 (12.6) **^,§ and^ ***^,#^	12 (7.2)
IgG positive, *n* (%)	18 (24.7)	216 (38.1) **^,§^	188 (35.7)	87 (37.7) *^,§^	66 (39.3) *^,§^
IgA positive, *n* (%)	20 (27.4)	156 (27.7)	129 (24.5)	57 (24.7)	48 (28.6)

A cut-off value for LtxA IgM positivity was defined using receiver operating characteristic (ROC) curves. The cut-off value for positivity for LtxA IgM was set at >64.7 AU/mL, giving a specificity of 98 %. PH, periodontitis healthy individuals; pre-sympt.i, pre-symptomatic individuals. ^§^ cases vs. PH, ^#^cases vs. controls, * *p* < 0.05, ** *p* < 0.01, *** *p* < 0.001.

**Table 2 jcm-09-01906-t002:** *Aa* exposure in study subjects according to anti-LtxA antibody isotype.

Anti-LtxA Isotype		Control	Pre-symptomatic Individuals			Early RA			Established RA		
IgM	IgG / IgA	*n* (%)	*n* (%)	* *p*	OR (95%CI)	*n* (%)	* *p*	OR (95%CI)	*n* (%)	* *p*	OR (95%CI)
-	Both (-)	330 (58.4)	312 (59.3)		1.0	128 (55.4)		**1.0**	96 (57.1)		**1.0**
-	Any (+)	214 (37.8)	181 (34.4)	0.406	0.90 (0.70, 1.16)	74 (32.0)	0.518	0.90 (0.64,1.25)	60 (35.7)	0.907	0.98 (0.68, 1.41)
+	Any (+)	11 (1.9)	24 (4.6)	0.039	2.12 (1.04, 4.30)	18 (7.8)	<0.001	3.88 (1.81, 8.26)	10 (6.0)	0.017	2.90 (1.21, 6.91)
+	Both (-)	11 (1.9)	9 (1.7)	0.751	0.87 (0.35, 2.12)	11 (4.8)	0.031	2.58 (1.09, 6.09)	2 (1.2)	0.935	0.95 (0.26, 3.47)
Total		566 (100)	526 (100)			231 (100)			168 (100)		

Combinations of anti-LtxA IgM and IgG/IgA positivity in pre-symptomatic individuals, early RA patients and established RA patients with established disease compared with controls (*n* = 566) as dependent variable analyzed in logistic regression analyses. * *p*-value for comparisons vs. Control. OR: Odds ratio, CI: Confidence interval.

**Table 3 jcm-09-01906-t003:** Anti-LtxA IgM levels (AU/mL) according to demographic factors.

		Pre-sympt.i. *n* = 526	* *p*	Early RA *n* = 231	* *p*	Established RA *n* = 168	* *p*	Control *n* = 73
**Sex**	Male	29.1 (4.48)	0.01	51.6 (10.49)	<0.01	16.4 (2.91)	ns	16.9 (1.79)
	Female	12.6 (2.23)	0.04	17.3 (2.77)	ns	22.0 (3)	<0.01	11.6 (1.15)
**Smoker**	Ever	19.3 (3.03)	ns	28.3 (3.48)	0.02	20.8 (2.77)	<0.01	14.9 (1.81)
	Never	14.0 (2.15)	ns	27.4 (9.92)	ns	16.8 (3.78)	ns	12.8 (1.2)
**RF**	Positive	25.9 (4.14)	ns	29.4 (4.82)	<0.01	21.5 (2.57)	ns	18.0 (3.68)
	Negative	13.4 (2.42)	ns	23.0 (5.23)	0.04	13.3 (3.68)	ns	13.2 (1.61)
**Anti-CCP**	Positive	18.9 (2.04)	ns	30.6 (4.85)	ns	20.7 (2.59)	ns	13.0 (3.38)
	Negative	16.4 (3.10)	ns	18.2 (3.92)	ns	18.3 (4.32)	ns	13.4 (1.01)
**HLA-SE**	Positive	17.9 (2.54)	0.03	24.3 (3.46)	<0.01	19.6 (2.75)	0.03	6.7 (2.17)
	Negative	17.5 (4.00)	ns	34.9 (9.85)	ns	21.2 (4.02)	<0.01	9.1 (3.27)

Pre-sympt.i., pre-symptomatic individuals; Anti-CCP, anti-cyclic-citrullinated peptide antibodies; RF, rheumatoid factor; HLA-SE, HLA-shared epitope; * vs. HC, Values are represented as mean (SEM). ns, non-significant (*p* > 0.05).

**Table 4 jcm-09-01906-t004:** Predictive value of anti-LtxA IgM levels for development of RA.

		Pre-symptomatic individuals				
	Unadjusted OR (95%CI)	Adjusted OR(95%CI)				
Anti-LtxA IgM, AU/mL	1.003	1.002	1.002	1.001	1.017	1.004
	(0.999, 1.007)	(0.998, 1.006)	(0.999, 1.006)	(0.997, 1.005)	(1.001, 1.034)	(1.000, 1.008)
					*p* = 0.033	*p* = 0.061
Ever Smoker		2.465				
		(1.924, 3.159)				
Anti-CCP			14.508			
			(9.223, 22.821)			
RF				4.931		
				(3.085, 7.882)		
HLA-SE					1.662	
					(1.016, 2.721)	
Sex						1.294
						(0.996, 1.681)
		**Early RA**				
	**Unadjusted OR (95% CI)**	**Adjusted OR(95% CI)**				
Anti-LtxA IgM, AU/mL	1.012	1.010	1.008	1.009	1.029	1.013
	(1.007, 1.017)	(1.005, 1.015)	(1.000, 1.016)	(1.001, 1.017)	(1.011, 1.047)	(1.007, 1.018)
	*p* < 0.001	*p* < 0.001	*p* = 0.040	*p* = 0.028	*p* = 0.002	*p* < 0.001
Ever Smoker		3.004				
		(2.156, 4.185)				
Anti-CCP			84,010			
			(49.697, 142.014)			
RF				32.543		
				(19.125, 55.376)		
HLA-SE					1.896	
					(1.096, 3.280)	
Sex						1.300
						(0.925, 1.838)
		**Established RA**				
	**Unadjusted OR (95% CI)**	**Adjusted OR (95% CI)**				
Anti-LtxA IgM, AU/mL	1.010	1.007	1.008	1.003	1.034	1.010
	(1.003, 1.015)	(1.001, 1.014)	(0.999, 1.017)	(0.991, 1.014)	(1.013, 1.056)	(1.003, 1.016)
	*p* = 0.003	*p* = 0.033			*p* = 0.002	*p* = 0.003
Ever Smoker		2.911				
		(2.006, 4.224)				
Anti-CCP			74.661			
			(43.089, 129.366)			
RF				53.893		
				(29.361, 98.924)		
HLA-SE					1.725	
					(0.970, 3.069)	
Sex						0.997
						(0.693, 1.442)

OR, odds ratio; CI, confidence interval; anti-CCP, anti-cyclic-citrullinated peptide antibodies; RF, rheumatoid factor; HLA-SE, Human leukocyte antigen ‘shared epitope’. The OR were calculated using logistic regression analyses with the stage of RA development (i.e., pre-symptomatic individuals, early RA, and established RA) as dependent variable. Anti-LtxA IgM was entered as continuous variable in AU/mL.

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
