# Peer review of "Exposure to Aggregatibacter actinomycetemcomitans before Symptom Onset and the Risk of Evolving to Rheumatoid Arthritis"

_jcm, 2020, doi:10.3390/jcm9061906_

Round 1

Reviewer 1 Report

This manuscript explores the link between Aa exposure and RA development. The data suggest that Aa recent exposure associates with development of the symptomatic phase of the disease. 

The manuscript is concise, very well written and presented. The ressources are solid. 

I would suggest to add trajectory of anti-LtxA antibodies for the 213 subjects with early RA and samples available before symptoms as well as trajectory of anti-LtxA antibodies for the 118 with all matched samples. 

Please clarify for logistic regression  the unit of antiLxA IgM used for OR calculation to appreciate the effect size of the association (is it per UA/ml or after z transformation ?) as the OR is very low. 

Author Response

We appreciate the reviewer comments.

Response #1: The trajectory of anti-LtxA IgM antibodies has been added as new Figure S1 (Supplementary material, page 3), and commented in the results section (lines 168-171). Trajectories for anti-LtxA IgG and IgA were not included because the data is negative.

Response #2: For the logistic regression in Table 4, anti-LtxA IgM levels were entered as an independent variable in AU/mL, which means that each AU/mL of anti-LtxA IgM contributes to an increase of 1.01 in the OR to develop early RA (i.e. a level of anti-LtxA IgM of 70 AU/mL equals an OR of 70.1 to develop early RA). This information has been clarified in the figure legend of Table 4 and in the discussion section, lines 254-256.

Reviewer 2 Report

This is a useful study which adds to our knowledge on connection between oral microflora and autoimmune diseases. Specifically, it potentially identifies a new role  of LtxA,  as a trigger of rheumatoid arthritis.  

  • - a Gram-negative bacterium
  • - "leukotoxin A" is not a name that usually used for LtxA, I would suggest using leukotoxin or RTX-toxin to avoid confusion

Assuming significant protein homology between different RTX-toxins I am wondering whether there is a possibility of cross reactivity. For instance 9D4 antibody against CyaA can recognize LtxA and K. kingae RtxA.  Have you performed any tests indicating that sera antibody are specific to LtxA only?

Author Response

We appreciate the reviewer comments,

Response #1: “A Gram-negative bacteria” was changed for “A Gram-negative bacterium” (Line 43).

Response #2: The reviewer is correct. However, changing the term of “leukotoxin A” for this manuscript may create confusion regarding other publications in the same topic. In particular, the use of “anti-LtxA antibodies” has been used to describe the association between Aa and RA. Even if we include an explanation for the change in the name, it is likely that some may consider that we are detecting a different type of antibody. Thus, with all respect, we wonder whether we could kept the use of LtxA for the manuscript.

Response #3: About possible cross-reactivity among RTX-proteins, nothing is reported for LtxA in the literature. In support to the specificity of anti-LtxA antibodies, previous studies have reported an exact match between the presence of Aa in the oral cavity with systemic anti-LtxA antibodies (J Periodont Res 2011; 46: 170–175, reference 17 in the manuscript). Moreover, Anders Johansson has tested possible cross-reactivity of LtxA antibodies in positive sera against E. coli, HlyA, and B. pertussis adenylate cyclase without any signs of cross reactivity. Indeed, the sequence homology between pore-forming proteins from the RTX-family is rather low (< 45%). Thus, while we cannot completely discard that LtxA may cross-react with some RTX-toxin that has not been tested, we consider that this possibility is quite low.

Round 2

Reviewer 1 Report

thank you for those clarifications. You meant a 70% increase (OR 1,7) I guess.